# Characterization of oral swab samples for diagnosis of pulmonary tuberculosis

Rachel C. Wood[1][ORCID], Alfred Andama[2,3][ORCID], Gleda Hermansky[4], Stephen Burkot[5¤a], Lucy Asege[2], Mukwatamundu Job[2], David Katumba[2], Martha Nakaye[2], Sandra Z. Mwebe[2], Jerry Mulondo[2], Christine M. Bachman[5¤a], Kevin P. Nichols[4¤a], Anne-Laure M. Le Ny[4¤a], Corrie Ortega[4¤a], Rita N. Olson[1], Kris M. Weigel[1], Alaina M. Olson[1], Damian Madan[4¤a], David Bell[5¤b], Adithya Cattamanchi[ORCID][6], William Worodria[2,3], Fred C. Semitala[2,3], Akos Somoskovi[5¤c], Gerard A. Cangelosi[1]*, Kyle J. Minch[ORCID][4¤a]

1 Department of Environmental and Occupational Health Sciences, School of Public Health, University of Washington, Seattle, Washington, United States of America, 2 Infectious Diseases Research Collaboration, Kampala, Uganda, 3 Department of Medicine, Makerere University College of Health Sciences, Kampala, Uganda, 4 Intellectual Ventures Laboratory, Bellevue, Washington, United States of America, 5 Intellectual Ventures' Global Good Fund, Bellevue, Washington, United States of America, 6 Division of Pulmonary and Critical Care Medicine and Center for Tuberculosis, Zuckerberg San Francisco General Hospital, University of California San Francisco, San Francisco, California, United States of America

[ORCID] These authors contributed equally to this work.
¤a Current address: GH Labs, Bellevue, Washington, United States of America.
¤b Current address: Independent Consultant, Issaquah, Washington, United States of America.
¤c Current address: Roche Molecular Systems, Inc., Pleasanton, California, United States of America.
* gcang@uw.edu

**Data Availability Statement:** All relevant data are within the paper and its Supporting Information files.

## Abstract

Oral swab analysis (OSA) has been shown to detect *Mycobacterium tuberculosis* (MTB) DNA in patients with pulmonary tuberculosis (TB). In previous analyses, qPCR testing of swab samples collected from tongue dorsa was up to 93% sensitive relative to sputum GeneXpert, when 2 swabs per patient were tested. The present study modified sample collection methods to increase sample biomass and characterized the viability of bacilli present in tongue swabs. A qPCR targeting conserved bacterial ribosomal rRNA gene (rDNA) sequences was used to quantify bacterial biomass in samples. There was no detectable reduction in total bacterial rDNA signal over the course of 10 rapidly repeated tongue samplings, indicating that swabs collect only a small portion of the biomass available for testing. Copan FLOQSwabs collected ~2-fold more biomass than Puritan PurFlock swabs, the best brand used previously (p = 0.006). FLOQSwabs were therefore evaluated in patients with possible TB in Uganda. A FLOQSwab was collected from each patient upon enrollment (Day 1) and, in a subset of sputum GeneXpert Ultra-positive patients, a second swab was collected on the following day (Day 2). Swabs were tested for MTB DNA by manual IS6110-targeted qPCR. Relative to sputum GeneXpert Ultra, single-swab sensitivity was 88% (44/50) on Day 1 and 94.4% (17/18) on Day 2. Specificity was 79.2% (42/53). Among an expanded sample of Ugandan patients, 62% (87/141) had colony-forming bacilli in their tongue dorsum swab samples. These findings will help guide further development of this promising TB screening method.

**Funding:** Study funding provided by The Global Good Fund I, LLC (www.globalgood.com), and by grants from the Bill & Melinda Gates Foundation (INV-004527) and the National Institute of Allergy and Infectious Diseases (R01AI139254). The funders had no role in study design, data collection and analysis, decision to publish, or preparation of the manuscript. At the time of study design and evaluation SB, CMB, DB, and AS received salary support from the Global Good Fund. At the time of study design and evaluation GH, KPN, AML, CO, DM, and KJM were employed by Intellectual Ventures Laboratory. The specific roles of these authors are articulated in the 'author contributions' section.

**Competing interests:** At the time of study design and evaluation SB, CMB, DB, and AS received salary support from the Global Good Fund. At the time of study design and evaluation GH, KPN, AML, CO, DM, and KJM were employed by Intellectual Ventures Laboratory. There are no patents, products in development or marketed products to declare. This does not alter our adherence to PLOS ONE policies on sharing data and materials. A number of the authors are currently affiliated with GH Labs and Roche Molecular Systems, but were not affiliated with them during the study, and these companies had no role in study design, data collection and analysis, decision to publish, or preparation of the manuscript.

## Introduction

Tuberculosis disease (TB), caused by *Mycobacterium tuberculosis* (MTB), remains a major global cause of morbidity and mortality [1]. The standard sample for TB diagnosis is sputum, a viscous material derived from patient airways. Sputum collection presents safety risks to health personnel, and the material is notably difficult to standardize and process for detection of MTB DNA. Sputum can be difficult for some patients to produce, especially children and those who are HIV-infected. The availability of alternative, noninvasive samples, which can easily be collected outside of the clinic, would increase the efficiency of testing and reduce the exposure risk to health care professionals [2, 3].

We and others have shown that MTB DNA is deposited on the oral epithelium during active TB disease and can be detected by oral swab analysis (OSA) [4–9]. In OSA, the dorsum of the tongue is gently brushed with a sterile disposable swab. The swab head with collected material, consisting of bacterial biofilm and host cells, is deposited into a sample buffer and eluted as a non-viscous suspension suitable for nucleic acid amplification testing (NAAT) targeting MTB DNA. Tongue swabbing is fast, painless, and does not require accommodations for privacy or aerosol control. Sputum-scarce patients such as children and HIV-positive adults are easily swabbed in any setting, and self-sampling is straightforward [10]. Therefore, OSA may be especially useful for TB case finding in non-clinical and community settings.

Studies on OSA for TB diagnosis have shown mixed results. In a blinded study conducted on 219 adult TB patients in South Africa, OSA exhibited 93% sensitivity and 92% specificity relative to sputum GeneXpert testing [4]. In a blinded study conducted on 201 children in South Africa, OSA matched or exceeded the diagnostic yield of induced sputum testing [6]. However, studies that used different methods from ours yielded more modest sensitivity values for OSA both in adults and children [7–9]. Moreover, in our South African studies, we collected two to three samples from each patient on separate days. Sensitivity and specificity were scored by calling a patient positive if either swab was positive [4–6]. A requirement for collecting and testing of multiple swabs per patient over multiple days would greatly reduce the utility of this approach.

In order to improve OSA-based TB testing, there is a need to optimize swab collection methods and to more fully understand the nature of the DNA biomarker at this anatomical site (for example, whether it is associated with viable MTB cells, with nonviable cells, or in cell-free form). Therefore, we conducted studies to: 1) compare the amounts of oral bacterial biomass collected by commercially-available swab products using a bacterial biomass proxy (qPCR measurement of conserved bacterial rDNA); 2) evaluate the accuracy of OSA-based TB PCR testing using the best oral swab product; and 3) assess whether viable MTB can be isolated and cultured from oral swab samples.

## Methods

### Study sites, populations, and sampling workflows

Healthy control participants in Seattle, King County, WA provided swabs for a rapid repeat swabbing analysis and swab comparison. For the rapid repeat analysis, 4 volunteers provided 10 swabs each with a maximum of 15 seconds between each swab collection. For the swab comparison, 3 volunteers each provided 5 swabs of each type.

For evaluations of sensitivity and specificity of OSA relative to sputum testing and of MTB viability in oral swab samples, we used a nested strategy as shown in Fig 1. Between October 2018 and April 2019 (6 months), patients with presumed pulmonary TB at Kiruddu Referral Hospital, Mulago National Referral Hospital TB ward, and Kisenyi Health Center IV out-

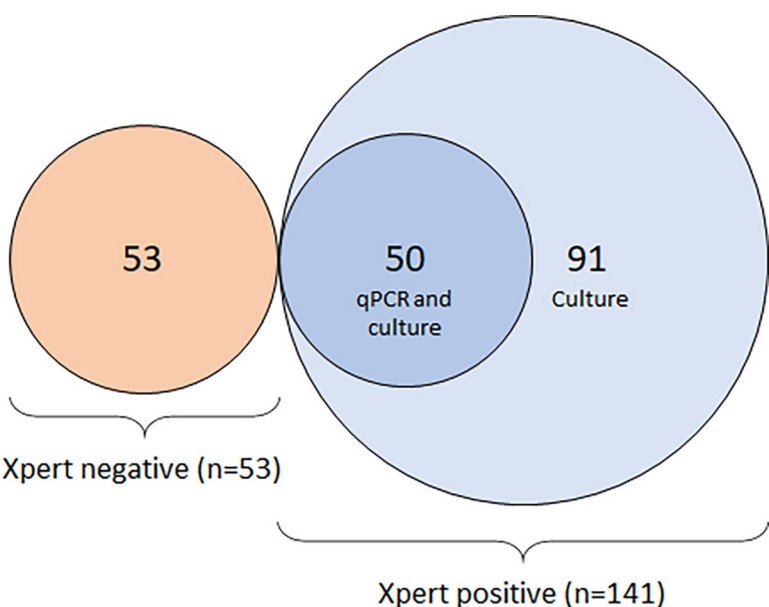

**Fig 1. Patient numbers for oral swab study.** Patients were enrolled in a nested strategy for MTB detection by qPCR (50 Xpert positive individuals and 53 Xpert negative individuals) or mycobacterial culture (141 Xpert positive individuals). All Xpert positive individuals from the qPCR arm were included in the culture arm.

patient clinic in Kampala, Uganda, were enrolled as previously described [11, 12]. We included adults (>18 years) who presented with respiratory symptoms and excluded participants who received anti-TB treatment or antibiotics with anti-TB activity such as fluoroquinolones in the prior 12 months or who refused or were unable to provide informed consent. After consent, all eligible participants completed a survey on demographics and medical history. Each patient provided at least 1 swab sample on Day 1 (n = 194 swabs from 144 patients, combining the sensitivity, specificity, and viability studies as shown in Fig 1). A subset of patients who were sputum GeneXpert positive were asked to return to provide a second sample on Day 2 (n = 41 swabs combining the sensitivity and viability studies). We matched enrollment of the first 50 GeneXpert positive individuals with 53 GeneXpert negative individuals. To assess viability of MTB cells in oral swab samples, an additional 91 GeneXpert positive individuals were enrolled (Fig 1).

The study workflow is outlined in S1 Fig. To accommodate assessment of sensitivity, specificity, and viability, the sampling protocol was as follows: a sputum sample was collected and split into two portions: the first portion was processed for Xpert (Xpert MTB/RIF Ultra assay, Cepheid, Sunnyvale, CA, USA), and the second portion was processed for sputum culture only. Immediately after collecting sputum for Xpert testing, an oral swab (for culture) was collected and processed for culture. After waiting for ≥ 1 hour, an oral swab (for PCR) was collected. A second sputum sample was collected and processed for culture, and 2 additional oral swabs were collected. Two swabs were collected from patients who returned on day 2 (n = 41)–the first swab was processed for culture, and the second swab was processed for PCR. These swabs were collected in the absence of preliminary prompted sputum production.

The study and full protocol were reviewed and approved by the Makerere University School of Medicine Research and Ethics Committee, the Uganda National Council for Science and Technology, the University of California San Francisco Committee on Human Research, and the Human Subjects Division of the University of Washington (STUDY00001840). The study

was performed according to the Standards for Reporting of Diagnostic Accuracy Studies (STARD) guidelines [13].

## Swab collection

Participants were asked to refrain from eating, drinking, brushing teeth, and using mouthwash for at least 30 minutes prior to swab collection. For the rapid repeat swabbing analysis, the Puritan PurFlock Ultra® (25-3606-U) was used. This swab type was evaluated in a previous study [4]. For the swab comparison, two swab products were evaluated: Puritan PurFlock Ultra swabs and Copan FLOQSwabs® (520CS01). Study staff asked subjects to stick out their tongue and using a sterile, individually wrapped swab, study staff ran the swab over the length and breadth of the front 2/3 of the subject's tongue. Pressure was sufficient to slightly bend the swab shafts. Samples were collected for 15–20 seconds while rotating the swab throughout.

## Sputum sample processing and analysis

Trained study technologists at Mulago National Referral Hospital Tuberculosis Laboratory and the Makerere University Mycobacteriology Laboratory performed all TB testing using standard protocols [14]. For Xpert, Cepheid sample reagent was added to the sputum sample at a 2:1 ratio. The mixture was vortexed for 10–15 seconds and incubated for 15 minutes at room temperature. Two milliliters (2 mL) of the liquefied sample were then transferred to the Xpert cartridge for testing in a four-module GeneXpert instrument using manufacturer standard settings.

Mycobacterial culture of sputum was performed by trained staff blinded to Xpert results. Sputum specimens were digested and decontaminated using standard methods [14]. Briefly, samples were treated with an equal volume of sodium hydroxide (Griffchem, cat #: 1310-73-2) and N-acetyl-Cysteine (SIGMA-Aldrich, cat #: 616-91-1) mixture to a final concentration of 1.5% for 15 minutes, neutralized with twice the volume of sterile phosphate-buffered solution pH 6.8, and centrifuged for 20 minutes at 3000 x g. The resulting sediment was resuspended in 2 mL of sterile phosphate-buffered solution at pH 6.8. Lowenstein-Jensen (BD, cat #: 220909) slant and Mycobacterial Growth Indicator Tube (MGIT) media were inoculated with 0.5 mL suspension, and Ziehl-Neelson (ZN) staining for acid-fast bacillus (AFB) smear microscopy was performed following standard protocols [14]. Prior to inoculation, a standard cocktail of antibiotics containing PANTA (SIGMA Aldrich: polymyxin B, CAS #: 1405-20-5; amphotericin B, CAS #: 1397-89-3; nalidixic acid solubilized with trimethoprim, CAS #: 23256-42-0; and azlocillin, CAS # 37091-65-9) mixed with 10% OADC (oleic acid, bovine serum albumin, dextrose, and catalase: BD, cat #: 212240) was added to MGIT tubes to suppress bacterial contamination and provide an optimum growth medium for mycobacteria. MGIT tubes were incubated in a BACTEC MGIT 960 instrument (BD; Franklin Lakes, NJ, USA) for up to 42 days and LJ slants were incubated at 37˚C for up to 56 days.

## Swab processing for mycobacteriological culture

Tongue swab samples, which are rich in fast-growing oral microorganisms, were decontaminated and cultured as follows. Working stocks of OMNIgene-SPUTUM (DNA Genotek, cat #: OM-SPD-250) were prepared every 30 days during the study. Stock solution (supplied at 2.0x stock concentration) was diluted 1:20 or 1:8 in sterile deionized water to create 0.1x or 0.25x working solutions, which were distributed in 1.0 mL aliquots into 2.0 mL gasketed screwcap tubes and stored protected from light at room temperature for 30 days. After 30 days, any unused aliquots were discarded, and fresh working solutions were prepared.

After tongue swab sample collection, swab heads were inserted into a screwcap tube containing OMNIgene-SPUTUM and rotated against sidewalls of tube for ~15 seconds. Swab heads were secured in capped tube and inverted 10 times to cover all surfaces with buffer. Swab samples were incubated at 25°C overnight ($\geq$ 18 hours, depending on time of collection) in OMNIgene-SPUTUM. Samples were pulse spun 2–3 seconds in a single-speed tabletop microfuge (Qor Labs 10,000 rpm Mini Centrifuge). For mycobacteriological culture, samples were taken from the swab/OMNIgene-SPUTUM tube (1.0 mL total volume) and inoculated directly to MGIT+PANTA (750 μL inoculum), LJ (100 μL inoculum), or 7H10+PACT (SIGMA Aldrich, cat #: 262710; polymyxin B, amphotericin B, carbenicillin, and trimethoprim; 100 μL inoculum). All cultures were incubated at 37°C and monitored for growth for up to 56 days. The identity of growth-positive cultures was confirmed acid-fast by ZN staining and to the MTB complex level by SD Bioline immunoassay to detect MTB antigen MPT64 following manufacturer-supplied methods (SD MPT64TB Ag kit, South Korea). Samples that remained negative after 56 days were discarded.

## Swab processing for qPCR

After collection, the head of the swab was inserted into a 2 mL tube containing 500 μL of a sterile lysis buffer (65 mM Tris pH 8.0, 50 mM EDTA, 50 mM sucrose, 100 mM NaCl, and 0.3% SDS) and snapped off. All swabs were stored at -80°C within 8 hours of collection.

The samples collected in Seattle and used for the rapid repeat analysis and the swab comparison were stored at -80°C and extracted using the QIAGEN QIAamp DNA mini kit. The samples were eluted into 300 μL (2 x 150 μL) of Buffer AE and stored at -20°C. To prepare for qPCR analysis targeting conserved bacterial rDNA, the samples were diluted 1:100.

The samples collected in Uganda were transported on dry ice to the Cangelosi lab in Seattle, WA. The laboratory team was blinded to the TB status of the Day 1 samples, though not the Day 2 samples (which were known to be from Xpert-positive participants). Before starting the extraction, each sample was split in half, and one half was stored at -80°C. The reserved half of the sample was kept as a precaution in case complications arose during the subsequent extraction and analysis. DNA was extracted using the QIAGEN QIAamp DNA mini kit, as described previously [4]. After elution into 300 μL of QIAGEN Buffer AE, 5 μL was used for qPCR analysis. For any sample that tested negative, 150 μL of the sample elution was ethanol precipitated [4, 5], resuspended in 15 μL of 3:1 molecular-grade H$_2$O and Buffer AE, and retested. The ethanol precipitation served to concentrate the sample allowing for qPCR analysis of a greater proportion of the sample.

## Swab sample analysis by qPCR

A qPCR assay targeting a conserved bacterial 16S rDNA sequence was used to compare the swab types based on the relative quantity of bacterial biomass that they gathered. The primer pair Com1/769R was used to amplify a 270 bp amplicon [15]. Each 20 μL reaction consisted of 1 μL of each 20 μM primer, 10 μL of iTaq Universal SYBR Green Supermix from BioRad, 6 μL molecular-grade H$_2$O, and 2 μL 1:100 diluted template. The cycling conditions were as described [15].

The qPCR analysis used for the detection of MTB in the samples collected in Kampala was described previously, for both the unconcentrated and concentrated samples [4, 5]. The reaction targets IS6110, a multicopy insertion element unique to the *M. tuberculosis* complex. Each qPCR run included a positive control containing a known amount of cultured MTB strain H37Ra extracted alongside the samples, a negative control consisting of sterile buffer extracted

with the samples, negative template PCR controls, and a standard curve made with purified H37Ra DNA.

## Results

### Characterization of biomass collection

Previously, we compared two swab brands for their abilities to detect MTB DNA on buccal (not tongue) surfaces in the mouths of adult TB patients. Based on Cq values from qPCR analysis, Puritan PurFlock Ultra swabs were found to collect about twice as much MTB DNA as Whatman OmniSwabs[R] (p = 0.015) [4]. This observation raised the possibility that signal strength (and therefore sensitivity) of OSA might be limited by the amount of biomass collected by some swab products, such that alternative products could enable greater sensitivity.

In order to test this possibility in the context of tongue swabbing without having to enroll new TB patients, we used normal oral flora as a measure of bacteria collected from the tongue dorsum surface. Biomass of collected bacteria was estimated by using a pan-bacterial domain qPCR that detects conserved portions of bacterial small subunit rDNA. While no single primer set is truly universal within the domain Bacteria, the broad-spectrum primer set Com1/769R has sufficient breadth to compare non-specific bacterial loads in paired analyses [12].

The first question we asked was whether bacterial biomass in samples is depleted over the course of repeated samplings. Four healthy US participants were repeatedly sampled with Puritan PurFlock Ultra swabs 10 times in rapid succession (approximately 10 seconds between each sampling). If tongue surface biomass was depleted, then we expected to see diminishing signals (rising Cq values) over the course of repeated sampling and testing by conserved bacterial rDNA qPCR; however, no such depletion was observed, and these data suggest that these swabs collect only a fraction of biomass that is available for sampling at this site (Fig 2, p = 0.099, one-way repeated measures ANOVA).

### An alternative swab brand collects more bacterial biomass

Next, we asked whether an alternative swab brand collects more bacterial biomass from the tongue dorsum than the best product identified previously, Puritan PurFlock Ultra [4]. Three healthy US volunteers each provided 5 samples using Puritan PurFlock Ultra or Copan FLOQSwabs. Collected bacterial biomass was quantified again by conserved bacterial rDNA qPCR as in Fig 2. Copan FLOQSwabs collected 2-fold more bacterial biomass than PurFlock Ultra (Fig 3). A paired t-test showed that this difference in Cq value was significant (p = 0.0064).

### Clinical evaluation of alternative swabs

Based on their biomass capacity, Copan FLOQSwabs were selected for evaluation in a clinical study in Kampala, Uganda. Adult patients with suspected active pulmonary TB (N = 191) were identified and enrolled by the study staff, with samples collected as described in Methods and S1 Fig. Socio-demographic and clinical characteristics of this cohort are summarized in Tables 1 and 2.

Of the 191 participants, 103 were included in the qPCR analysis, as shown in Fig 1. Of these, 50 (48.5%) tested positive for TB by sputum GeneXpert MTB/RIF Ultra, and 47 (45.6%) tested positive for TB by sputum culture. Negative results were obtained for 53 (51.5%) and 56 (54.4%) of these participants by Xpert and sputum culture, respectively. As described in Methods, swabs were collected after sputum collection on Day 1. Of the 50 sputum Xpert-positive individuals enrolled on Day 1, 18 returned on Day 2 for oral swab sampling without prior

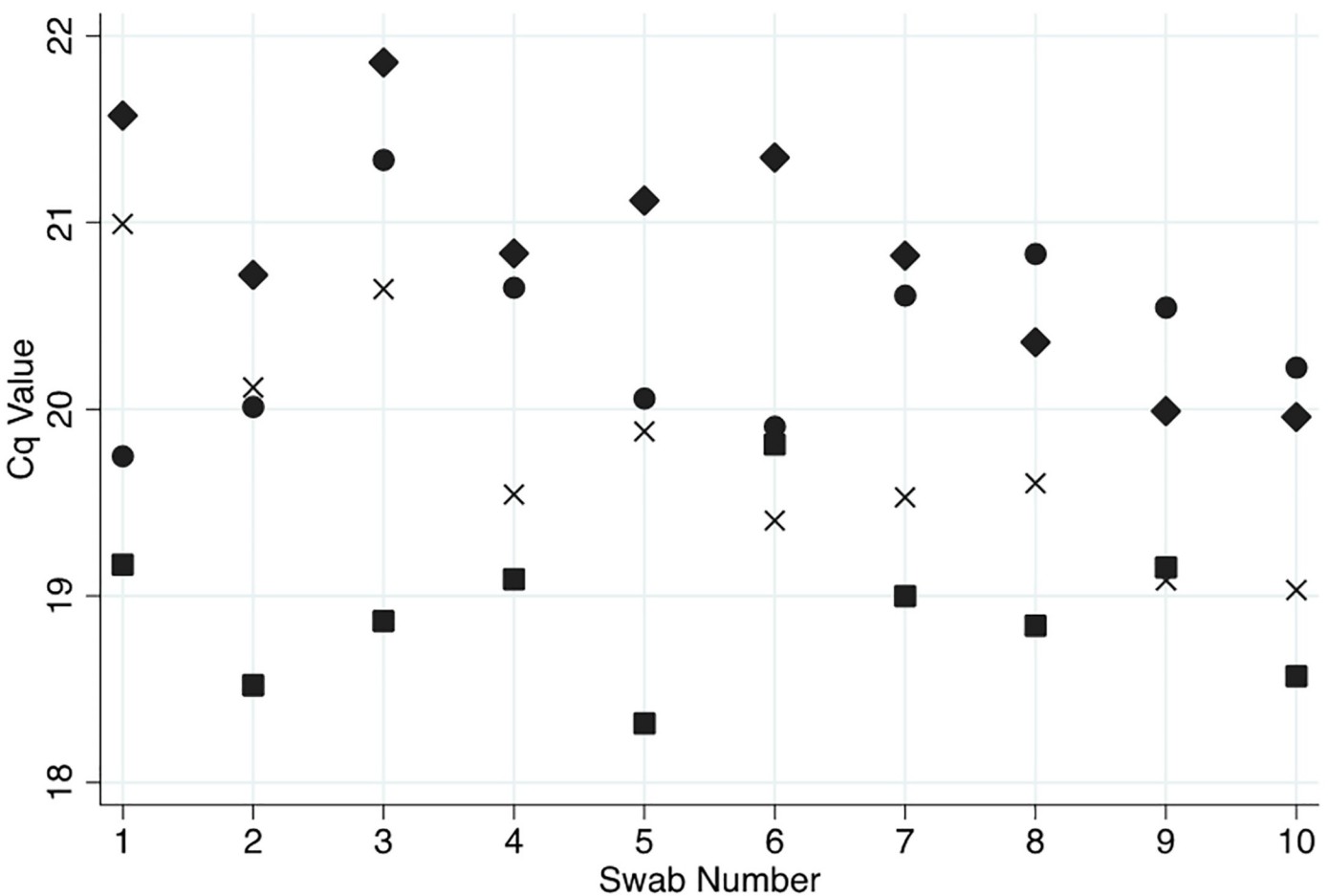

**Fig 2. Rapid repeat sampling of individual participants.** Ten samples were collected in rapid succession from the tongue dorsa of four healthy volunteers, and tested by conserved bacterial rDNA qPCR. Each individual is represented by a distinct type of symbol.

prompted sputum production. Two of these subjects subsequently delivered negative results by sputum culture. In contrast to previous studies [4, 5], here we measured sensitivity of OSA based on single swab results, rather than multiple swabs. Day 1 swabs exhibited 88.0% (44/50) and 91.5% (43/47) sensitivity relative to sputum Xpert and culture results, respectively (Table 3). Specificity of Day 1 swabs was 79.2% (42/53) and 66.1% (37/56) relative to sputum Xpert and culture, respectively. Day 2 swabs exhibited 94.4% (17/18) and 93.3% (15/16) sensitivity relative to sputum Xpert and culture, respectively (Table 3). The two sputum culture-negative Day 2 subjects who were positive by sputum GeneXpert were also positive by OSA.

## Factors associated with OSA results

No significant associations were observed between OSA positivity and smoking, previous TB infection, gender, having a household contact with TB, clinical site (Kiruddu Hospital vs. Kisenyi Health Centre), or patient type (in-patient vs. out-patient). However, among patients with positive oral swabs, an association was observed between OSA signal strength and HIV co-infection. Swabs from patients co-infected with HIV had higher Cq values, which indicates a weaker signal (Table 4).

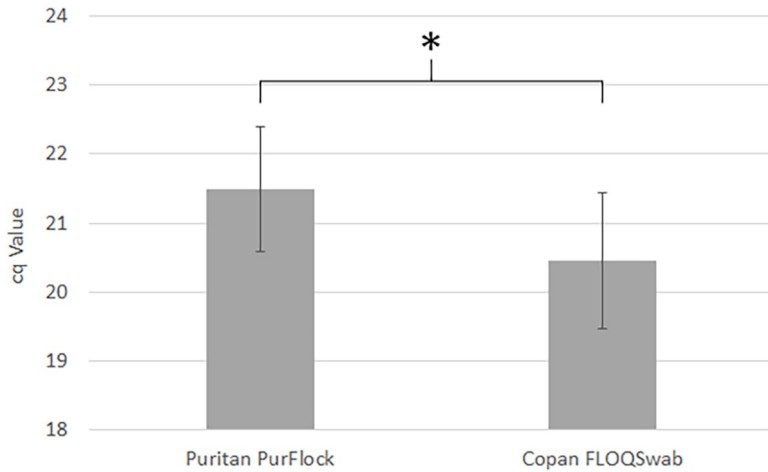

**Fig 3. Comparison of total bacterial biomass collected from tongue dorsa by two swab products.** Five samples were collected from 3 healthy volunteers using Puritan Purflock swabs or Copan FLOQSwabs. Bacterial biomass was quantified by qPCR using primers against a conserved bacterial rDNA locus. Bars represent mean Cq +/- standard deviation. Significance calculated using paired t-test (p = 0.0064).

## Viability of MTB cells in tongue swab samples

We hypothesized that some fraction of TB patients harbor viable MTB cells that can be isolated, and cultured, from the oral cavity. To test this hypothesis, we analyzed samples from all 141 participants with sputum Xpert-confirmed TB (Fig 1). We designed this study to assess sensitivity, rather than specificity, of the oral swab for culture approach, and, accordingly, we enrolled only patients with positive sputum Xpert results.

Swab samples were collected, processed, and cultured for viable MTB as described in Methods and S1 Fig. Colonies were identified to the species level. At least one Day 1, post-sputum oral swab culture was positive in 82 of 141 (58.2%, Table 5) GeneXpert-positive patients. Of the 41 patients who returned on Day 2 for an additional swab collection without prior

**Table 1. Socio-demographic characteristics and other patient information.**

|  | TB patients (Xpert or culture-positive) (n = 142) | Non-TB patients (Xpert and culture-negative) (n = 52) | p-value |
|---|---|---|---|
| *Age in years (%)* |  |  |  |
| Mean (SD) | 32.1 (9.6) | 34.2 (11) |  |
| Median (IQR) | 30 (13.8) | 34 (14) |  |
| *Gender (%)* |  |  |  |
| Female | 46 (32.4) | 24 (46.1) | 0.08 |
| Male | 96 (67.6) | 28 (53.9) | 0.08 |
| *Smoker (%)* | 21 (14.8) |  |  |
| *Enrollment Site (%)* | n = 141 |  |  |
| Kiruddu | 19 (13.5) | 16 (30.8) | 0.006* |
| Kisenyi | 95 (67.4) | 36 (69.2) | 0.8 |
| Mulago | 27 (19.1) | 0 (0) |  |
| *Patient type (%)* |  |  |  |
| In-patient | 8 (5.6) | 3 (5.8) | 1 |
| Out-patient | 134 (94.4) | 49 (94.2) | 0.8 |

*Significant at p < 0.05, z-score for 2 population proportions.

**Table 2. Clinical characteristic of patients.**

| | TB patients (Xpert or culture-positive) (n = 142) | Non-TB patients (Xpert and culture-negative) (n = 52) | p-value |
|---|---|---|---|
| *Current cough (%)* | 142 (100) | 52 (100) | < 0.0001* |
| *Duration of current cough (days)* | | | |
| Mean (SD) | 70.9 (52.9) | 43.8 (57) | |
| Median (IQR) | 30 (60) | 25.5 (46) | |
| *Coughing up blood (%)* | 38 (26.8) | 10 (19.2) | 0.3 |
| *Fever (%)* | 137 (96.5) | 47 (90.4) | 0.09 |
| *Night sweats (%)* | 135 (95) | 40 (76.9) | 0.0002* |
| *Weight loss (%)* | 139 (98) | 46 (88.5) | 0.006* |
| *Weight loss > 5kg (%)* | 118 (85), (n = 139) | 24 (52.2), (n = 46) | < 0.0001* |
| *Decreased appetite (%)* | 128 (91.4), (n = 140) | 39 (75) | 0.003* |
| *Swollen lymph nodes (%)* | 12 (8.5), (n = 141) | 2 (3.9) | 0.3 |
| *Stomach pain or swelling (%)* | 21 (15), (n = 140) | 7 (13.5) | 0.8 |
| *Oxygen saturation (%)* | | | |
| Mean (SD) | 96.5 (2.8) | 97.7 (2.7) | |
| Median (IQR) | 97 (2) | 98 (1) | |
| *ECOG performance score (%)* | | | |
| 0 | 9 (6.3) | 8 (15.4) | 0.05* |
| 1 | 63 (44.4) | 30 (57.7) | 0.1 |
| 2 | 65 (45.8) | 13 (25) | 0.009* |
| 3 | 5 (3.5) | 1 (1.9) | 0.6 |
| *HIV infection (%)* | 36 (25.4) | 19 (36.5) | 0.1 |
| *Household TB contact (%)* | 10 (7) | 1 (1.9) | 0.2 |
| *Previous TB disease (%)* | 10 (7) | 4 (7.7) | 0.9 |

*Significant at p < 0.05, z-score for 2 population proportions.

prompted sputum production, oral swab cultures were positive in 18 (43.9%, Table 5). This proportion increased to 50.0% (18 of 36) when excluding patients in whom all oral swab cultures were contaminated.

While the collection design for Days 1 and 2 differed with respect to preceding sputum production, 87 of 141 (61.7%) patients had MTB-positive tongue swab cultures when considering all samples together. Patients with higher sputum smear grades or higher semi-quantitative sputum Xpert results were more likely to have cultivatable MTB from tongue swabs (Table 6; Pearson correlation coefficient for smear grade = 0.875, and Xpert semi-quantitative result = 0.983). Tongue swab cultures were positive for MTB in 17 of 35 (48.6%) HIV co-infected individuals, and this was not significantly different compared to HIV negative individuals (p > 0.1, Fisher's Exact Test).

## Discussion

Oral swabs have many potential advantages for TB screening and triage, especially in community settings where the collection of sputum or urine isn't practical. However, evaluations of

**Table 3. Sensitivity and specificity of OSA relative to sputum GeneXpert Ultra and culture.**

| | Day 1 swabs | | Day 2 swabs | |
|---|---|---|---|---|
| | Sensitivity | Specificity | Sensitivity | Specificity |
| Relative to Xpert | 44/50 (88%) | 42/53 (79.2%) | 17/18 (94.4%) | N/A[1] |
| Relative to culture | 43/47 (91.5%) | 37/56 (66.1%) | 15/16 (93.8%) | 0/2 (0%) |

[1]Not applicable. All Day 2 subjects were Xpert-positive.

the method have yielded mixed indications of sensitivity [4–8]. In some studies, multiple swabs had to be tested to yield sensitivity values above 90% relative to sputum GeneXpert or culture [4, 5]. Oral swab samples are small in volume and unlikely to contain large numbers of bacilli in all cases. Therefore, we evaluated alternative sample collection methods designed to increase sample biomass.

Our laboratory analysis using conserved bacterial rDNA indicated that a previously-used tongue swabbing method collected only a small fraction of the bacterial biomass that exists on this surface. Assuming that MTB cells at this site are entrained in the tongue dorsum biofilm or the underlying epithelium, it seems feasible that sensitivity can be improved by increasing the amount of collected material. Through use of the bacterial biomass proxy, we identified the Copan FLOQSwab as a product that collects more biomass than the products used previously.

The clinical evaluation in Kampala yielded promising results with this product, albeit within a small sample set. Where previously it had taken 2 swabs per patient to achieve up to 93% sensitivity in adults, the current study exhibited up to 94% sensitivity relative to sputum culture using just one swab per patient. This was a different population from previous studies, and we did not do side-by-side clinical comparisons between the two swab brands. However, Copan FLOQSwabs performed well in this analysis and, we now use it exclusively for OSA.

Several limitations should be noted. First, even with increased biomass collection, sensitivity was less than 100% relative to both GeneXpert and culture. False-negative swabs were tested quantitatively for human DNA by using a sample adequacy control described recently [16]. They appeared to have been properly collected. Therefore, some patients who are sputum-positive can be missed by OSA. Second, specificity at 79% and 66% relative to sputum Xpert and culture, respectively, was markedly lower than the 92% observed previously [4]. Negative controls did not indicate laboratory contamination, so higher levels of false positivity in the present study may have had other causes. One possibility is that high-capacity FLOQSwabs are better able to collect MTB DNA that may be present in the oral cavities of people without TB disease in high-prevalence environments. Third, samples were tested by using manual qPCR. Automated methodologies are needed that exploit the specific advantages of swab samples relative to sputum. Fourth, this study was embedded within a larger study whose properties required the collection of Day 1 swabs after on-site collection of sputum. Production of sputum prior to swab sampling could have affected swab results by depositing fresh sputum onto the tongue. However, Day 2 swabs were collected without prior sputum production, and there

**Table 4. Association between OSA signal strength and HIV co-infection among patients with positive tongue swabs.**

| | HIV co-infected | HIV non-infected | p-value (t-test) |
|---|---|---|---|
| | Cq ± SD (N) | Cq ± SD (N) | |
| Day 1 swabs | 34.35 ± 4.1 (9) | 31.88 ± 2.9 (30) | 0.047 |
| Day 2 swabs | 35.77 ± 4.2 (7) | 31.09 ± 1.8 (8) | 0.012 |

**Table 5. Tuberculosis culture positivity among tongue swab samples from Xpert positive individuals.**

|  | MTB positive/total (%) |
|---|---|
| **Sputum culture** | 132/141 (93.6) |
| **First Day 1 swab** | 71/141 (50.4) |
| **Second Day 1 swab** | 61/141 (43.3) |
| **Day 1 combined (either or both positive)** | 82/141 (58.2) |
| **Day 2 swab** | 18/41 (43.9) |
| **Any swab positive on Day 1 or 2** | 87/141 (61.7) |
| **HIV+ individuals (any swab positive)** | 17/35 (48.6) |

was no evidence for reduced sensitivity under these conditions. Fifth, although Day 1 swab analyses were blinded with regard to TB status, Day 2 analyses were not blinded due to the study's design. Finally, the sample size was small.

Despite these limitations, the results show promise in OSA as an easy-to-collect, noninvasive sample for TB screening and diagnosis. If larger studies continue to exhibit sensitivity in excess of 90% using a single swab, the method has the potential to meet at least some of the criteria needed for community-based triage testing [17].

Swabs from TB patients co-infected with HIV yielded higher Cq values (weaker signals) on average than those who were not co-infected with HIV. This confirms and extends an observation reported previously in a South African population [4].

The results from swab culture experiments supported the hypothesis that viable MTB can be cultured from a swab of the tongue dorsum, with sensitivity relative to sputum culture of 58% and 50% for swabs collected following sputum or in the absence of prompted sputum production, respectively. As with the molecular analysis, a limiting factor was that patients provided sputum before Day 1 swab sampling, which poses the risk of artificially loading the tongue dorsum with MTB. However, some patients returned and provided Day 2 swabs without prior sputum collection, and there was no significant difference in results between Day 1 and Day 2 swabs (P > 0.1, Fisher's exact test). This analysis suggests that the detection of viable cells was not an artifact of study design. The results inform our understanding of OSA by confirming that at least part of the signal is associated with whole, viable MTB cells. This has implications for occupational safety of the method in addition to the further development of methods for sample handling and processing. These findings may also affect our understanding of the dynamics of TB transmission.

**Table 6. Correlation between swab culture and smear microscopy grade or GeneXpert semiquantitative result.**

|  |  | MTB culture positive/total for category, n = 87, (%) |
|---|---|---|
| GeneXpert Semiquantitative Result | **Trace** | 1/4 (25.0) |
|  | **Very Low** | 5/13 (38.5) |
|  | **Low** | 16/30 (53.3) |
|  | **Medium** | 28/40 (70.0) |
|  | **High** | 37/47 (78.7) |
| Smear Microscopy Grade | **Negative** | 14/33 (42.4) |
|  | **Scanty** | 6/14 (42.9) |
|  | **1+** | 24/31 (77.4) |
|  | **2+** | 32/42 (76.2) |
|  | **3+** | 11/14 (78.6) |

In summary, we used swab culture to improve our understanding of the physiology of MTB present on the tongue dorsum and a bacterial biomass proxy to identify a product with increased capacity to collect dorsum biofilm for MTB testing. The product performed well in a clinical assessment, exhibiting single-swab sensitivity in a range that approaches the needs of triage testing. Furthermore, the results in a Ugandan population confirmed and extended previous findings from the study in South Africa [4, 5]. With continued improvement, OSA could become an effective noninvasive, non-sputum sampling method for TB diagnosis and screening.

## Supporting information

**S1 File. Source table containing data informing all figures, tables, and analyses described in study.**
(CSV)

**S1 Fig. Study design for sample collection and processing.**
(TIF)

## Acknowledgments

We wish to thank Santina Castriciano of Copan Italia for providing swabs and technical guidance.

## Author Contributions

**Conceptualization:** Kevin P. Nichols, Corrie Ortega, Damian Madan, David Bell, Adithya Cattamanchi, Akos Somoskovi, Gerard A. Cangelosi, Kyle J. Minch.

**Data curation:** Alfred Andama, Stephen Burkot, Kris M. Weigel.

**Formal analysis:** Alfred Andama, Stephen Burkot, Kyle J. Minch.

**Funding acquisition:** David Bell, Akos Somoskovi.

**Investigation:** Rachel C. Wood, Alfred Andama, Gleda Hermansky, Lucy Asege, Mukwatamundu Job, David Katumba, Martha Nakaye, Sandra Z. Mwebe, Jerry Mulondo, Corrie Ortega, Rita N. Olson, Kris M. Weigel, Alaina M. Olson, William Worodria, Kyle J. Minch.

**Methodology:** Rachel C. Wood, Gleda Hermansky, Christine M. Bachman, Rita N. Olson, Kris M. Weigel, Alaina M. Olson, Kyle J. Minch.

**Project administration:** Christine M. Bachman, Corrie Ortega, Akos Somoskovi, Gerard A. Cangelosi.

**Resources:** David Bell.

**Supervision:** Alfred Andama, Jerry Mulondo, Kevin P. Nichols, Anne-Laure M. Le Ny, Damian Madan, Adithya Cattamanchi, Fred C. Semitala, Akos Somoskovi, Gerard A. Cangelosi, Kyle J. Minch.

**Validation:** Rachel C. Wood, Alfred Andama, Gleda Hermansky, Kyle J. Minch.

**Visualization:** Stephen Burkot, Kyle J. Minch.

**Writing – original draft:** Rachel C. Wood, Alfred Andama, Adithya Cattamanchi, Gerard A. Cangelosi, Kyle J. Minch.

**Writing – review & editing:** Rachel C. Wood, Alfred Andama, Gleda Hermansky, Stephen Burkot, Lucy Asege, Mukwatamundu Job, David Katumba, Martha Nakaye, Sandra Z. Mwebe, Jerry Mulondo, Christine M. Bachman, Kevin P. Nichols, Anne-Laure M. Le Ny, Corrie Ortega, Rita N. Olson, Kris M. Weigel, Alaina M. Olson, Damian Madan, David Bell, Adithya Cattamanchi, William Worodria, Fred C. Semitala, Akos Somoskovi, Gerard A. Cangelosi, Kyle J. Minch.

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
