## [Decision Letter · Decision Letter 0]

9 Oct 2020

PONE-D-20-28429

Viable *Mycobacterium tuberculosis* from swabs of the tongue dorsum of pulmonary tuberculosis patients

PLOS ONE

Dear Dr. Minch,

Thank you for submitting your manuscript to PLOS ONE. After careful consideration, we feel that it has merit but does not fully meet PLOS ONE’s publication criteria as it currently stands. Therefore, we invite you to submit a revised version of the manuscript that addresses the points raised during the review process.

The authors should pay attention to the critiques raised by the reviewers. All reviewers feel that the study is too simplistic and could be improved for a better reach of audience. Particularly, modifying the abstract (Reviewer#1) and other details (Reviewer#2) are important. Regarding the latter, I suggest the authors to add more data to improve the standard of their manuscript and the reach/impact of this methodology to the community. We can provide additional time if the authors are willing to perform additional experiments.

We look forward to receiving your revised manuscript.

Kind regards,

Selvakumar Subbian, Ph.D.

Academic Editor

PLOS ONE

Journal Requirements:

'Study funding provided by The Global Good Fund I, LLC (www.globalgood.com). The funders had no role in study design, data collection and analysis, decision to publish, or preparation of the manuscript.'

We note that one or more of the authors are employed by commercial companies:

Intellectual Ventures Laboratory, Global Health Labs and Roche Molecular Systems, Inc.

a. Please provide an amended Funding Statement declaring these commercial affiliations, as well as a statement regarding the Role of Funders in your study. If the funding organization did not play a role in the study design, data collection and analysis, decision to publish, or preparation of the manuscript and only provided financial support in the form of authors' salaries and/or research materials, please review your statements relating to the author contributions, and ensure you have specifically and accurately indicated the role(s) that these authors had in your study. You can update author roles in the Author Contributions section of the online submission form.

b. Please also provide an updated Competing Interests Statement declaring these commercial affiliations along with any other relevant declarations relating to employment, consultancy, patents, products in development, or marketed products, etc.  

Reviewers' comments:

Reviewer's Responses to Questions

**Comments to the Author**

1. Is the manuscript technically sound, and do the data support the conclusions?

Reviewer #1: Partly

Reviewer #2: Yes

Reviewer #3: Partly

Reviewer #4: Partly

2. Has the statistical analysis been performed appropriately and rigorously? 

Reviewer #1: Yes

Reviewer #2: Yes

Reviewer #3: Yes

Reviewer #4: No

3. Have the authors made all data underlying the findings in their manuscript fully available?

Reviewer #1: Yes

Reviewer #2: No

Reviewer #3: No

Reviewer #4: Yes

4. Is the manuscript presented in an intelligible fashion and written in standard English?

Reviewer #1: Yes

Reviewer #2: Yes

Reviewer #3: Yes

Reviewer #4: Yes

5. Review Comments to the Author

Reviewer #1: Abstract: “Sensitivities of the collection methods are not sufficient for use in routine culture based diagnostic tesating” is one of the main findings, but the abstract gives a different view, and conveys the message that this collection procedure will be an alternative to the rest stating that it might facilitate culture based diagnostic testing for TB. This needs to be rephrased in a manner so that the reader might get a true picture.

The positive findings of the study related to the comparison of the results with molecular based techniques are all reported previously and the abstract reiterates the same. The only positive finding is that viable bacilli is present in tongue, but this is expected and the authors have tried to merely substantiate this. But this message is not seen in the abstract.

Materials and methods: (Line 110 ) It is mentioned that all the cultures were incubated for 56 days, But is it the same for MGIT systems also?

Results and Discussion: (Line 133) It is not clear if the figure 61.4% includes single oral swab culture results or both, do specify this.

Table 3 legend shows swab culture, but is not reflected in the table title

(Line 163) This statement is contraindicative, says it is not statistically significant in the second half, but the sentence begins that the assessment is most discriminatory and how do you justify that the second day specimen is able to do so and the reason behind this.

Line 169 is confusing and explain what does oral sampling rate stands for.

Reviewer #2: This is an interesting but a simplistic and short study. In this reviewer's opinion it does not reach the level of a research article - a short communication would be more appropriate. The following are to be noted:

1. There is no clear description of the clinical application of the oral swab method.

2. No clear description/implications or insights for the understanding of TB pathogenesis presented/discussed.

3. The authors are respectfully advised to substantially expand the scope of the paper for it to fly as a full fledged research article.

4. On p3, lines 51-54 the tongue swab procedure is described stating three swabs were taken. The first and the third swabs are mentioned clearly but when was the 2nd one taken? It is not clear.

5. Table 1 shows Day2 culture positivity for the one swab sample taken is 43.9%. The authors have not discussed why they think it is so much lower than the first swab on Day 1 (50.4).

Reviewer #3: The manuscript “Viable Mycobacterium tuberculosis from swabs of the tongue dorsum of

pulmonary tuberculosis patients” is quite interesting and informative.

The paper is simple and concise but few more additions seems imperative before it can be considered for publication. The study emphasizes on the use of non-sputum method in the context of use in paucibacillary situation as an alternate to sputum. There are few clarifications needed.

1. Apart from culturing of MTB, was there any molecular diagnostic test performed on these tongue swabs? The gene Xpert results used for selection criteria is performed on the sputum samples and not the tongue swabs. If this method is going to be suggested as an alternate to sputum collection a parallel rapid diagnostic test to first identify positive and then its co-relation to their growth as culture will be more valid.

2. The bacilli were cultured in MGIT, LJ and 7H10. It would be very useful to give a comprehensive table in place of table 2 given currently to understand the revival and contamination rate in different MTB media assessed here.

3. It would be interesting to add the drug resistance data on the isolates and any indications to cultivability.

Reviewer #4: The authors in this manuscript describe an approach for isolation of viable Mycobacterium tuberculosis from swabs of the tongue dorsum of pulmonary tuberculosis patients. The study looks interesting, but the authors should address the following concerns.

• The results indicate that viable mtb bacilli do exist in the tongue dorsum, but the culture positivity is quite low. How will it improve the current TB diagnosis portfolio? Where do the authors want to position this test? Molecular tests are gaining advantage and use of DNA isolated from these samples as an input in endorsed molecular tests will provide a platform for positioning this technique.

• It will be useful to see the contamination rates of OminiGene vs. the NALC-NaOH method for the sputum versus swabs, as the former is reported to provide lower contamination rates, but here the rates were quite high (130/477 were contaminated). It will be useful to standardize the decontamination procedure for tongue swabs.

• Was the sample stored after reconstitution in OMNIgene tube? Did that make a difference in culture positivity? What advantage does the author foresee for this method? It will be interesting to see the viability assessment of bacilli stored in the OMNIgene tube over a period of time.

• Line 192. What bio-safety measures do the authors suggest, in view of their results. Also, a follow-up experiment might be useful to see how does the mtb load decrease in the dorsum during the course of therapy.

• What was interesting was the number of samples positive by swab culture (42.4%) in smear-ve category; what category of Xpert result did these samples lie in? A head to head comparison of Xpert versus Smear grade in all samples might be useful.

6. PLOS authors have the option to publish the peer review history of their article (what does this mean?). If published, this will include your full peer review and any attached files.

Reviewer #1: **Yes: **Azger Dusthackeer

Reviewer #2: No

Reviewer #3: No

Reviewer #4: No

---

## [Author Response · Author response to Decision Letter 0]

18 Feb 2021

PONE-D-20-28429 Manuscript Re-submission

Text included here is also contained within the associated "Response to Reviewers" document. We appreciate the reviewers’ thorough read-through and comments on our manuscript, and in the rewrite/resubmission process we paid particular attention to the consistent critiques that the study was overly would benefit from additional data and a revised scope. We have substantially updated the manuscript, and responded to individual reviewer comments, below. 

Point-by-point response to reviewer comments:

Reviewer #1: 

1. Abstract: “Sensitivities of the collection methods are not sufficient for use in routine culture based diagnostic testing” is one of the main findings, but the abstract gives a different view, and conveys the message that this collection procedure will be an alternative to the rest stating that it might facilitate culture based diagnostic testing for TB. This needs to be rephrased in a manner so that the reader might get a true picture.

The positive findings of the study related to the comparison of the results with molecular based techniques are all reported previously and the abstract reiterates the same. The only positive finding is that viable bacilli is present in tongue, but this is expected and the authors have tried to merely substantiate this. But this message is not seen in the abstract.

Response: Please review revised abstract in which we have focused more on the qPCR/molecular results that are now central to this manuscript. The information regarding culture positivity is included as an arm of the study, but without reference to using oral swab cultures for diagnosis.

2. Materials and methods: (Line 110 ) It is mentioned that all the cultures were incubated for 56 days, But is it the same for MGIT systems also?

Response: MGIT cultures were incubated for 42 days. We have revised the text to make this clear (Line 171).

3. Results and Discussion: (Line 133) It is not clear if the figure 61.4% includes single oral swab culture results or both, do specify this.

Response: The original text as written indicated “While the collection design for Days 1 and 2 differed with respect to preceding sputum production, 87 of 141 (61.7 %) patients had MTB-positive tongue swab cultures when considering all samples together.” We have revised this language to clarify: “While the collection design for Days 1 and 2 differed with respect to preceding sputum production, 87 of 141 (61.7 %) patients had MTB-positive tongue swab cultures when considering all samples together.” (Lines 335 – 337). 

4. Table 3 legend shows swab culture, but is not reflected in the table title

Response: In the revised manuscript these data are included as Table 6, and we have revised the table title to clarify based on this point: “Table 6. Correlation between swab culture and smear microscopy grade or GeneXpert semiquantitative result.”

5. (Line 163) This statement is contraindicative, says it is not statistically significant in the second half, but the sentence begins that the assessment is most discriminatory and how do you justify that the second day specimen is able to do so and the reason behind this.

Response: The first half of the sentence was intended a comment that the Day 2 samples (collected without prior prompted sputum production) are more likely to be reflective of a patient’s natural disease state, while the observation that there was not a significant difference between Day 2 (no prior prompted sputum) and Day 1 samples (prior prompted sputum) is a comment on study design. We have modified the language in the revised draft, which can be found on lines 396-400. 

6. Line 169 is confusing and explain what does oral sampling rate stands for.

Response: Thank you for raising this. This was a word omission on our part, and should have read “…oral sampling culture contamination rate…”. With the focus on the qPCR/molecular results in the revised manuscript we have removed this text.

Reviewer #2: This is an interesting but a simplistic and short study. In this reviewer's opinion it does not reach the level of a research article - a short communication would be more appropriate. The following are to be noted:

1. There is no clear description of the clinical application of the oral swab method.

Response: We have addressed this concern in the revised manuscript with a greater focus on the diagnostic potential of qPCR from oral swab sampling. The integration of this information can be found across the manuscript; however, we specifically point out Lines 63 – 81 in the Introduction, the sociodemographic and clinical characteristics of the patients included in the study (Tables 1 & 2), comparison to GeneXpert of swab-based molecular MTB diagnosis (Tables 3 & 4), the text throughout the Results section associated with these tables, and the expanded text in the Discussion (for example, Lines 362 – 367 and Lines 406 – 413).

2. No clear description/implications or insights for the understanding of TB pathogenesis presented/discussed.

Response: In the revised manuscript we have shifted the emphasis from insights on TB pathogenesis attendant with the phenomenon of culturable MTB on the tongue dorsum, to a molecular diagnostic focus. We believe the culture/viability data still support the observation that knowledge of viable MTB on the tongue contributes to our knowledge base of the pathogen, but as the reviewer points out in the absence of additional data we limit further speculation on TB pathogenesis.

3. The authors are respectfully advised to substantially expand the scope of the paper for it to fly as a full fledged research article.

Response: Thank you for this comment and suggestion. In line with other reviewers and editor suggestion, we hope that our revised manuscript and expanded scope satisfy this request.

4. On p3, lines 51-54 the tongue swab procedure is described stating three swabs were taken. The first and the third swabs are mentioned clearly but when was the 2nd one taken? It is not clear.

Response: We have added a supplemental figure, “Fig S1” with a complete workflow of sample collection.

5. Table 1 shows Day2 culture positivity for the one swab sample taken is 43.9%. The authors have not discussed why they think it is so much lower than the first swab on Day 1 (50.4).

Response: The information raised in this comment can now be found in Table 5 and is described in the text on Lines 335 – 336 and Lines 393 – 401, in which we suggest that while it there is not a statistically significant difference in culture tongue swab culture positivity with or without preceding sputum production, it is possible that this procedural detail in the sample collection workflow (see Figure S1) explains the discrepancy. 

Reviewer #3: The manuscript “Viable Mycobacterium tuberculosis from swabs of the tongue dorsum of

pulmonary tuberculosis patients” is quite interesting and informative.

The paper is simple and concise but few more additions seems imperative before it can be considered for publication. The study emphasizes on the use of non-sputum method in the context of use in paucibacillary situation as an alternate to sputum. There are few clarifications needed.

1. Apart from culturing of MTB, was there any molecular diagnostic test performed on these tongue swabs? The gene Xpert results used for selection criteria is performed on the sputum samples and not the tongue swabs. If this method is going to be suggested as an alternate to sputum collection a parallel rapid diagnostic test to first identify positive and then its co-relation to their growth as culture will be more valid.

Response: Consistent with comments from other reviewers, we have substantially revised the manuscript to include molecular/qPCR results to address this concern.

2. The bacilli were cultured in MGIT, LJ and 7H10. It would be very useful to give a comprehensive table in place of table 2 given currently to understand the revival and contamination rate in different MTB media assessed here.

Response: With the revised focus on molecular detection of MTB from tongue swab, we felt that the content of the former table 2 distracted from the new focus. We have removed this table from the manuscript.

3. It would be interesting to add the drug resistance data on the isolates and any indications to cultivability.

Response: We agree that these are interesting questions; however, in our study setting, focused drug resistance (as determined by Xpert Rif analysis) is quite low (2 of 141 individuals, 1.4 %), and with the sample size it is difficult to draw significant conclusions on this question. This may be an excellent subject for follow-up studies, perhaps with a larger sample size and/or in settings with a greater burden of drug resistant MTB.

Reviewer #4: The authors in this manuscript describe an approach for isolation of viable Mycobacterium tuberculosis from swabs of the tongue dorsum of pulmonary tuberculosis patients. The study looks interesting, but the authors should address the following concerns.

1. The results indicate that viable mtb bacilli do exist in the tongue dorsum, but the culture positivity is quite low. How will it improve the current TB diagnosis portfolio? Where do the authors want to position this test? Molecular tests are gaining advantage and use of DNA isolated from these samples as an input in endorsed molecular tests will provide a platform for positioning this technique.

Response: We appreciate this comment and agree with the reviewer’s assessment that molecular tests are gaining advantage in MTB diagnosis. Our revised manuscript is updated accordingly. We have retained data regarding viability of MTB collected with tongue swabs (Tables 5 & 6, with associated text Lines 319 – 346), however our focus is on the operational rather than diagnostic implications of these data (see, for example, Lines 401 – 405).

2. It will be useful to see the contamination rates of OminiGene vs. the NALC-NaOH method for the sputum versus swabs, as the former is reported to provide lower contamination rates, but here the rates were quite high (130/477 were contaminated). It will be useful to standardize the decontamination procedure for tongue swabs.

Response: While we retain our data on the viability of MTB from tongue swabs, our revised manuscript focuses on the ongoing efforts to develop (and ultimately standardize) oral swab analysis for molecular detection of MTB. We agree that if any tongue swab method were to be adopted for routine use, standardization of methodsis a critical component. 

3. Was the sample stored after reconstitution in OMNIgene tube? Did that make a difference in culture positivity? What advantage does the author foresee for this method? It will be interesting to see the viability assessment of bacilli stored in the OMNIgene tube over a period of time.

Response: We have clarified the description of the methods to address this point (see Materials & Methods, Lines 183 - 196). The swab samples were stored in OMNIgene-SPUTUM for 18-24 hours at 25 °C prior to sample processing and culture inoculation.

4. Line 192. What bio-safety measures do the authors suggest, in view of their results. Also, a follow-up experiment might be useful to see how does the mtb load decrease in the dorsum during the course of therapy.

Response: While a data-driven answer to the question of MTB load over the course of therapy was outside the design and scope of this analysis, we agree that this is an interesting question for follow up work, as it may offer insights into the efficacy of a given therapy particularly in cases of drug resistance. While we were unable to conduct those experiments with our patient cohort we hypothesize that oral swab culture positivity rates over the course of treatment would track with sputum culture positivity rates. 

5. What was interesting was the number of samples positive by swab culture (42.4%) in smear-ve category; what category of Xpert result did these samples lie in? A head to head comparison of Xpert versus Smear grade in all samples might be useful.

Response: We agree that these data reinforce the observation that MTB culture is more sensitive than smear microscopy, and can now point to tongue swab as an additional sampling matrix that follows this pattern; however, we feel that a head-to-head comparison of sputum Xpert results versus sputum smear microscopy falls outside of the scope of the current manuscript. We note, here, that for the majority of individuals (8 of 14) from this smear negative group had “very low” or “low” Xpert results, and the remainder (6 of 14) were split evenly between medium and high Xpert results.

---

## [Decision Letter · Decision Letter 1]

11 Apr 2021

PONE-D-20-28429R1

Characterization of oral swab samples for diagnosis of pulmonary tuberculosis

PLOS ONE

Dear Dr. Minch,

Thank you for submitting your manuscript to PLOS ONE. After careful consideration, we feel that it has merit but does not fully meet PLOS ONE’s publication criteria as it currently stands. Therefore, we invite you to submit a revised version of the manuscript that addresses the points raised during the review process.

ACADEMIC EDITOR: Although two of the reviewers are contended with the revised manuscript, one of the reviewers has raised some minor issues that needs to be addressed by a revision. I encourage the authors to take this opportunity to check for any other issues, such as typos and grammatical errors.

We look forward to receiving your revised manuscript.

Kind regards,

Selvakumar Subbian, Ph.D.

Academic Editor

PLOS ONE

Journal Requirements:

Reviewers' comments:

Reviewer's Responses to Questions

**Comments to the Author**

1. If the authors have adequately addressed your comments raised in a previous round of review and you feel that this manuscript is now acceptable for publication, you may indicate that here to bypass the “Comments to the Author” section, enter your conflict of interest statement in the “Confidential to Editor” section, and submit your "Accept" recommendation.

Reviewer #1: All comments have been addressed

Reviewer #3: (No Response)

Reviewer #4: All comments have been addressed

2. Is the manuscript technically sound, and do the data support the conclusions?

Reviewer #1: Yes

Reviewer #3: Partly

Reviewer #4: Yes

3. Has the statistical analysis been performed appropriately and rigorously? 

Reviewer #1: Yes

Reviewer #3: Yes

Reviewer #4: Yes

4. Have the authors made all data underlying the findings in their manuscript fully available?

Reviewer #1: Yes

Reviewer #3: Yes

Reviewer #4: Yes

5. Is the manuscript presented in an intelligible fashion and written in standard English?

Reviewer #1: Yes

Reviewer #3: No

Reviewer #4: Yes

6. Review Comments to the Author

Reviewer #1: The manuscript could be accepted in the present form. Authors have addressed the queries raised and the response is acceptable and the edits have been made as per their replies.

Reviewer #3: The paper titled “Characterization of oral swab samples for diagnosis of pulmonary tuberculosis” is a neat study. In the present form it lacks being presented as a full-length article only based on methodology and its results. However, with the variety of papers that characterise oral swabs for diagnosis published in last 5 years, the authors need to justify how this study is different and how this study is a value addition to the previous ones. The scope of the paper in comparison to previously published ones is still lacking. The results need to be discussed in detail in context of previous published studies and not just the opinion of the authors about the same in the present context. There is no insight on the recommendations for culture of mycobacteria and preferred method for maximum recovery of viable bacteria among the three.

Line 49 Make it uniform % (87/114) to make it easy to understand.

Line 36 refers to geneXpert while line 45, 47 and 288 refers to GeneXpert Ultra, please give correct detail. Is it MTB/RIF or Ultra? Throughout the manuscript it reads MTB/Rif – eg Line 120 as the first portion was processed for Xpert (Xpert MTB/RIF assay, Cepheid, Sunnyvale, CA, USA).The authors got to be specific.

Line 51 – Referring to oral swabs as a screening in asymptomatic patients? While line 63 says “oral epithelium during 64 active TB disease” Line 71 says “useful for diagnosis and screening in non-clinical and community settings” Please clarify

Line 101 adults (>18 years) – please provide interquartile range for this.

Line 190-192 Clarify - the incubation periods for LJ and MGIT or 7H10 are not the same.

Line 238-240 - Based on Cq values from qPCR 239 analysis, Puritan PurFlock Ultra swabs were found to collect about twice as much MTB DNA as Whatman OmniSwabs® 240 (p = 0.015). Kindly discuss this in comparison to previous reported studies.

Line 365 This was a different population from previous studies, 366 and we did not do side-by-side clinical comparisons between the two swab brands. Please add further discussion to the swab type variation in comparison to previous studies.

Fig S1 refers to MGIT or LJ or both. Please confirm reference to 56 days wait for confirming culture negative.

Reviewer #4: The manuscript has improved in clarity from the previous version. The authors have addressed all the comments in the revised manuscript.

7. PLOS authors have the option to publish the peer review history of their article (what does this mean?). If published, this will include your full peer review and any attached files.

Reviewer #1: No

Reviewer #3: No

Reviewer #4: No

---

## [Author Response · Author response to Decision Letter 1]

23 Apr 2021

Point-by-point response to reviewer comments

NOTE: Line numbers in our responses refer to the mark-up version. Major changes are also highlighted in the mark-up version.

Reviewer #3: 

1. The paper titled “Characterization of oral swab samples for diagnosis of pulmonary tuberculosis” is a neat study. In the present form it lacks being presented as a full-length article only based on methodology and its results. However, with the variety of papers that characterise oral swabs for diagnosis published in last 5 years, the authors need to justify how this study is different and how this study is a value addition to the previous ones. The scope of the paper in comparison to previously published ones is still lacking. The results need to be discussed in detail in context of previous published studies and not just the opinion of the authors about the same in the present context. There is no insight on the recommendations for culture of mycobacteria and preferred method for maximum recovery of viable bacteria among the three.

o We thank the reviewer for this summary of critiques, and point to several areas in the manuscript where their concerns are addressed. We note that key differences that distinguish this work from previous studies are articulated in the abstract (Lines 38-41, highlighted in the Track Changes version): “In previous analyses, qPCR testing of swab samples collected from tongue dorsa was up to 93 % sensitive relative to sputum GeneXpert, when 2 swabs per patient were tested. The present study modified sample collection methods to increase sample biomass and characterized the viability of bacilli present in tongue swabs.” We note further that we discuss this study/these results in comparison to previous work in several other locations in the manuscript (see, for example, lines 88-95 and in several highlighted passages in the Discussion). 

o The reviewer also flags that there is no insight on the recommendation for culture of mycobacteria. As discussed in the cover letter associated with the manuscript resubmission, following the reviews of our original submission, we substantially revised the manuscript to focus on the molecular detection of MTB from oral swabs, rather than broad diagnostic application of culture-based methods. With that revised focus, in the present manuscript we do make note of considerations for occupational safety, and suggest that the demonstration that at least some DNA collected from the tongue is associated with whole, viable MTB cells has implications for sample handling and processing (Lines 407-411).

2. Line 49 Make it uniform % (87/114) to make it easy to understand.

o The text has been updated. 

3. Line 36 refers to geneXpert while line 45, 47 and 288 refers to GeneXpert Ultra, please give correct detail. Is it MTB/RIF or Ultra? Throughout the manuscript it reads MTB/Rif – eg Line 120 as the first portion was processed for Xpert (Xpert MTB/RIF assay, Cepheid, Sunnyvale, CA, USA).The authors got to be specific.

o The comment on Line 36 refers to results published in a different study, and in that study the authors used sputum GeneXpert (as stated in the abstract). Within the current manuscript, we implemented the nomenclature suggested by the reviewer (GeneXpert MTB/Rif Ultra) throughout the manuscript. This information is also reflected in columns G and H in Table S1. For consistency, we have added the designator “Ultra” to line 126. 

4. Line 51 – Referring to oral swabs as a screening in asymptomatic patients? While line 63 says “oral epithelium during 64 active TB disease” Line 71 says “useful for diagnosis and screening in non-clinical and community settings” Please clarify

o Line 63 is a description of previously published data. In this study our study enrollment criteria were to include patients “…who presented with respiratory symptoms…” (Line 107), and so we do not make comment on the screening of asymptomatic patients in the abstract. To minimize confusion we changed the wording that the reviewer flags to: “useful for TB case finding in non-clinical and community settings” (Line 76).

5. Line 101 adults (>18 years) – please provide interquartile range for this.

o The requested data are included in Table 1. The text on Lines 96-104 is a description of our prospective study design and inclusion criteria. 

6. Line 190-192 Clarify - the incubation periods for LJ and MGIT or 7H10 are not the same.

o As described in Lines 198-199, and in Fig S1, in this study all cultures from tongue swab were incubated for up to 56 days. We acknowledge that guidance documents indicate different incubation periods for the various media, however, in this study investigating the tongue as a source matrix for viable MTB we adopted a conservative, long, incubation/observation period of 56 days uniform across all media types. As described in the section titled “Sputum sample processing and analysis,” for sputum culture, we incubated (sputum) MGIT cultures for up to 42 days and (sputum) LJ cultures for up to 56 days (lines 177-179).

7. Line 238-240 - Based on Cq values from qPCR 239 analysis, Puritan PurFlock Ultra swabs were found to collect about twice as much MTB DNA as Whatman OmniSwabs® 240 (p = 0.015). Kindly discuss this in comparison to previous reported studies.

o This sentence refers to results reported previously (reference 4, Luabeya et al 2019), as noted in the preceding sentence. The complete passage reads “Previously, we compared two swab brands for their abilities to detect MTB DNA on buccal (not tongue) surfaces in the mouths of adult TB patients. Based on Cq values from qPCR analysis, Puritan PurFlock Ultra swabs were found to collect about twice as much MTB DNA as Whatman OmniSwabs® (p = 0.015) [4].”

8. Line 365 This was a different population from previous studies, 366 and we did not do side-by-side clinical comparisons between the two swab brands. Please add further discussion to the swab type variation in comparison to previous studies.

o These comparisons and context are presented throughout the current manuscript. Please see, for example: 

The paragraph on Lines 77-88 

Lines 370-372 “…previously it had taken 2 swab per patient… the current study exhibited up to 94 % sensitivity relative to sputum culture using just one swab per patient.” 

9. Fig S1 refers to MGIT or LJ or both. Please confirm reference to 56 days wait for confirming culture negative.

o This comment is addressed in response #6, above.

---

## [Editor Report · Decision Letter 2]

27 Apr 2021

Characterization of oral swab samples for diagnosis of pulmonary tuberculosis

PONE-D-20-28429R2

Dear Dr. Cangelosi,

We’re pleased to inform you that your manuscript has been judged scientifically suitable for publication and will be formally accepted for publication once it meets all outstanding technical requirements.

Kind regards,

Selvakumar Subbian, Ph.D.

Academic Editor

PLOS ONE
---

## [Editor Report · Acceptance letter]

7 May 2021

PONE-D-20-28429R2 

Characterization of oral swab samples for diagnosis of pulmonary tuberculosis 

Dear Dr. Cangelosi:

I'm pleased to inform you that your manuscript has been deemed suitable for publication in PLOS ONE. Congratulations! Your manuscript is now with our production department. 

Kind regards, 

on behalf of

Dr. Selvakumar Subbian 

Academic Editor

PLOS ONE